# Application of Pseudo-Three-Dimensional Residual Network to Classify the Stages of Moyamoya Disease

**DOI:** 10.3390/brainsci13050742

**Published:** 2023-04-29

**Authors:** Jiawei Xu, Jie Wu, Yu Lei, Yuxiang Gu

**Affiliations:** 1School of Health Science & Engineering, University of Shanghai for Science & Technology, Shanghai 200093, China; 2Department of Neurosurgery, Huashan Hospital, Fudan University, Shanghai 200040, China

**Keywords:** moyamoya disease, Digital Subtraction Angiography, deep learning, dilated convolution, Pseudo-Three-Dimensional Residual Network, feature extraction

## Abstract

It is essential to assess the condition of moyamoya disease (MMD) patients accurately and promptly to prevent MMD from endangering their lives. A Pseudo-Three-Dimensional Residual Network (P3D ResNet) was proposed to process spatial and temporal information, which was implemented in the identification of MMD stages. Digital Subtraction Angiography (DSA) sequences were split into mild, moderate and severe stages in accordance with the progression of MMD, and divided into a training set, a verification set, and a test set with a ratio of 6:2:2 after data enhancement. The features of the DSA images were processed using decoupled three-dimensional (3D) convolution. To increase the receptive field and preserve the features of the vessels, decoupled 3D dilated convolutions that are equivalent to two-dimensional dilated convolutions, plus one-dimensional dilated convolution, were utilized in the spatial and temporal domains, respectively. Then, they were coupled in serial, parallel, and serial–parallel modes to form P3D modules based on the structure of the residual unit. The three kinds of module were placed in a proper sequence to create the complete P3D ResNet. The experimental results demonstrate that the accuracy of P3D ResNet can reach 95.78% with appropriate parameter quantities, making it easy to implement in a clinical setting.

## 1. Introduction

The cause of moyamoya disease (MMD), a relatively unusual cerebrovascular disease, is unknown. It is known as MMD because it is characterized by stenosis or occlusion in the terminal part of the internal carotid arteries (ICA), the beginning part of the middle cerebral artery (MCA) and the anterior cerebral artery (ACA), which results in the presence of small vessels that resemble smoke in the brain [1]. MMD is quite damaging, with a high mortality rate and disability rate. The clinical symptoms of MMD include ischemic and hemorrhagic strokes. MMD brought on by a hemorrhagic stroke will seriously damage the cranial nervous system, which is a significant factor in patients’ poor prognosis and eventual death [2,3,4]. Once a cerebrovascular accident occurs, it will cause permanent damage to the body, and may even cause the patient to die [5]. Magnetic Resonance Angiography (MRA), Computed Tomography Angiography (CTA), and Digital Subtraction Angiography (DSA) are the three most commonly utilized diagnostic methods for MMD. The gold standard for the diagnosis of MMD in clinical practice is DSA [6], and the diagnosis is based on pathological alterations in the cerebral vasculature that occur in MMD [7]. To create a continuous subtraction image sequence with a high temporal resolution and a high dynamic resolution, DSA equipment can continually take a few to dozens of images per second.

In 1969, Japanese researchers Suzuki and Takaku [8] created the Suzuki stage as a standard method for determining the extent of vasculopathy in MMD patients. According to the patient’s cerebral angiography, MMD can be classified into six stages: carotid fork narrowing, moyamoya initiation, moyamoya intensification, moyamoya minimization, moyamoya reduction, and moyamoya disappearance. Briefly, stages 1–2 are when moyamoya-like vessels first start to progressively form in the brain. Stages 3–4 are when the moyamoya-like vessels are increased. With the gradual elimination of moyamoya-like vessels, stages 5–6 are the stages of compensated vessel formation. The goal and greatest challenge of current research is remains the accurate classification of the stages of MMD and the prediction of MMD. Instead of using the precise Suzuki stage, Stages 1–2, stages 3–4, and stages 5–6 of MMD were combined into three grades: mild, moderate, and severe. First, there are many other factors that must be considered in addition to the severity of ICA lesions when determining the prognosis of MMD patients. The prognosis of MMD patients with ischemic stroke is considerably impacted by the compensatory capacity of collateral circulation [9]. The ischemic area is maintained by collateral or new vessels when severe ICA stenosis or occlusion occurs, preventing insufficient blood flow and minimizing brain tissue damage. Monitoring collateral circulation in the region of cerebral ischemia is crucial [10]. Because of this, it is more practical to categorize MMD into three stages in this study.

Deep Learning (DL) has gained significant traction in the field of intelligent medical treatment in recent years, and reliable research breakthroughs have been made in the automatic identification of MMD. Tackeun et al. [11] trained a neural network with six convolution layers to recognize MMD on CTA modality with 84.1% accuracy. The accuracy of the improved VGG16 network employed by Akiyama et al. [12] in diagnosing MMD was 92.8%. MMD identification frequently employs Convolutional Neural Networks (CNN), although it is unable to concurrently account for the spatio-temporal information in the sequence. Three-dimensional (3D) convolution is currently one of the mainstays for the simultaneous analysis of spatial and temporal data. In comparison to two-dimensional (2D) convolution, the temporal dimension is added to 3D convolution to process information between frames, which has led to some success in the investigation of behavior identification in videos [13,14,15,16]. Based on this, it successfully employs 3D convolution to identify MMD in a DSA sequence. In order to extract the long-term temporal and spatial features of a DSA picture sequence, Hu et al. [17] employed a 2D CNN and a Bidirectional Convolutional Gated Recurrent Unit (BiConvGRU), while the short-term temporal and spatial data were further extracted using a 3D CNN. Its accuracy, sensitivity, and specificity were 0.9788, 0.9780, and 0.9796. Automatic staging and precise prognosis can be achieved based on the automatic identification and detection of MMD. The spatio-temporal features in the video sequence can be effectively extracted using 3D convolution; however, this involves a high computing cost, requires the development of a new 3D CNN, and consumes large amounts of memory space, according to some studies [18,19,20]. Therefore, it is crucial to use better 3D convolution to tackle this issue [21].

In this paper, an automatic staging technique for MMD based on a Pseudo-3D (P3D) Residual Network was provided. First, P3D convolution kernels were defined to classify the stages of MMD automatically. These kernels processed spatial and temporal data separately using equivalent 2D convolution and one-dimensional (1D) convolution. Dilative convolution was employed to expand the receptive field without lowering the resolution ratio, which enabled the network to focus on multi-scale context information. Finally, P3D modules were created based on the residual unit to prevent gradient explosion and gradient disappearance induced by the rise in network depth. The 2D dilative convolution kernel and 1D dilative convolution kernel were combined in serial, parallel, and serial–parallel fashions, respectively, to form a P3D Residual Network (ResNet). A P3D ResNet realizes the automatic staging of MMD and provides a necessary reference for the prognosis of MMD. 

## 2. Materials and Methods

### 2.1. Data Processing

The department of neurosurgery, Huashan Hospital, Fudan University, provided the information used in this experiment. In total, we gathered 406 samples of MMD cases between July 2017 and October 2020. DSA images that were affected by intense artifacts were eliminated. All patients in our database were diagnosed independently by two senior neurosurgeons via routine procedures. If a consensus was not reached, the whole treatment team discussed the case together and came to a final consensus. We were able to obtain 137 mild, 412 moderate, and 174 severe MMD hemispheres. First, the starting frame was chosen to be the instant when the contrast medium had just passed through the end of the ICA, and the beginning of the ACA and MCA. The DSA sequence needed for this experiment was taken from this frame and the following 9 frames, for a total of 10 frames. Second, we extracted the region of interest (ROI), that is, the terminal part of the ICA, the MCA and the ACA, from the DSA sequence, and processed them as 224 × 224 pixels. As is shown in Figure 1, after the ROI was divided, the influence of the skull and other irrelevant parts was removed. We used the augmentation technique to address the issue of imbalance in the experimental data caused by the majority of the samples being moderate MMD. The DSA images were rotated and flipped throughout this process. In addition, we used test time augmentation [22,23], that is, enhancement of the data in the test set. The number of mild, moderate, and severe MMD hemisphere cases obtained was 516, 512, and 515, respectively. Last but not least, the data were split into training, validation, and test sets with a ratio of 6:2:2.

### 2.2. Operating Environment

In this experiment, we used a Nvidia Tesla V100 graphics card with 16GB video RAM and an Intel (R) Xeon (R) CPU e5-2640 V4 @ 2.40 GHz processor with 128GB memory. It was put into practice using the PyTorch DL framework in a Linux environment using anaconda3.7, cuda10.0, and python3.6.

### 2.3. Design of P3D ResNet

Neural networks have exceptional technological benefits in the area of image processing because they can extract image features through the convolution layer, learn the internal rules of data samples, and obtain the distributed feature representation of data. The retrieved feature information is richer the deeper the network becomes, and the network also performs better. However, gradient disappearance and gradient explosion are likely to happen if the model’s depth reaches its maximum. Additionally, a wide model leads to excessive parameters, a high risk of over-fitting, and difficulty in model optimization. The performance of deep networks is still not totally satisfying, although data initialization and regularization can stop the network training process from stagnating. He et al. [24] created a deep ResNet using identity mapping to continuously learn new features, considerably enhancing their model’s capacity to learn features. The prevailing consensus is that medical data sets typically consist of tiny sample sizes. We suggest using the infrastructure of ResNet to reduce the risk of over-fitting in small MMD sample data sets in deep networks, and breaking down the original 3 × 3 × 3 3D convolution kernel into 1 × 3 × 3 convolution kernels (equivalent to 2D CNN) for the spatial domain and 3 × 1 × 1 convolution kernels (equivalent to 1D CNN) for the temporal domain, in that order. Dilative convolution can also retain more feature details, expand the network’s receptive field while extracting features, and enhance generalization capacity when utilized in both the spatial and temporal domains. Three P3D modules, comprising a combination of serial, parallel, and serial–parallel 2D dilative convolution and 1D dilative convolution, were designed to maximize the effects of the two types of dilative convolution kernel. These three P3D modules, designated P3D-A, P3D-B, and P3D-C, were linked alternatively to replace the original residual unit in ResNet. Figure 2 depicts the structure of P3D ResNet. 

#### 2.3.1. Residual Unit

The identity map (arc part in Figure 3) and residual map (straight line part in Figure 3) are connected to the output by jump connections in each residual unit, as shown in Figure 3. This operation can produce the final output from the input data x and the output data F(x) obtained through the weight layer, that is, H(x) = F(x) + x. The residual mapping path passes through the weight layers, increasing the depth of the network and enhancing its functionality. It is customary to raise the network’s layer count in order to improve the network’s accuracy and feature extraction capabilities. The network will approach saturation once the number of layers reaches a certain level, and network degradation will become a problem. Due to the identity mapping that is applied to the residual unit, the ultimate result can still be H(x) = x, even if F(x) is 0. Without identity mapping, the network loses its capacity for forward propagation and back propagation, and the parameter update of weight layers becomes stalled, making it impossible for the network to learn new features. Additionally, leveraging jump connections to incorporate the input data into the output data can significantly increase the integrity of information, ease the burden of training the network, and lower computational costs.

#### 2.3.2. P3D Modules

The *T × S × S* 3D convolution kernel is decomposed into *1 × S × S* and *T × 1 × 1* convolution kernels, where *T* represents the temporal dimensional convolution parameters in the 3D convolution kernel and *S* represents the spatial convolution parameters in the 3D convolution kernel. This action is taken because the conventional 3D convolution kernel merges spatial and temporal information, which is detrimental to the optimization of the model. The spatial information and temporal information of DSA sequences are processed independently using the two convolution kernels through decomposition. P3D convolution is the name given to this type of decoupled 3D convolution. By breaking down the 3D convolution kernel in the spatio-temporal domain, it is possible to significantly lower the number of parameters and reduce the calculating cost. In addition, multiple nonlinear operations in the module make it more capable of learning features.

As can be seen in Figure 4, three P3D residual modules, named P3D-A, P3D-B, and P3D-C, were designed. Among them, P3D-A connects spatial convolution S and temporal convolution T in series, ensuring the depth of the network under the same receptive field conditions, and improving the performance of the network to a certain extent. P3D-B uses a parallel structure to facilitate the distributed computing of features. P3D-C integrates series and parallel operations into a module, effectively fusing and supplementing feature information, so as to enrich it.

(1)P3D-A: Both the spatial and temporal dimension convolutions are cascaded in P3D-A. To create the final result, the feature maps are first used to perform a 2D spatial convolution calculation, followed by a 1D temporal convolution calculation. The equation can be written as follows:


(1)
$$\left(I+T\cdot S\right)\cdot x_{t}≔x_{t}+T\left(S\left(x_{t}\right)\right)=x_{t+1}$$


(2)P3D-B: There is no symbiotic relationship between spatial and temporal dimension convolutions. The two run parallel to one another. These two outcomes can be combined with the input of the module to obtain the final output. The equation reads as follows:


(2)
$$\left(I+T+S\right)\cdot x_{t}≔x_{t}+S\left(x_{t}\right)+T\left(x_{t}\right)=x_{t+1}$$


(3)P3D-C: This operation combines the two earlier approaches. The input first passes through spatial 2D convolution, and the results are then added to those of the temporal 1D convolution operation. Finally, it is possible to establish the following formula:


(3)
$$\left(I+S+T\cdot S\right)\cdot x_{t}≔x_{t}+S\left(x_{t}\right)+T\left(S\left(x_{t}\right)\right)=x_{t+1}$$


Among these equations, $x_{t}$ is the input of the module, $x_{t+1}$ is the output of the module, T is the convolution of the temporal dimension, and S is the convolution of the spatial dimension.

#### 2.3.3. Dilated Convolution

The feature extraction process is improved by adding dilated convolution to the spatial and temporal dimensions of the P3D module, respectively, so that the details of vessels can be better preserved and the perception of context information can be enhanced. This action is taken to prevent the pooling operation in the network from reducing the resolution of the feature map, and to better determine the dependency between frames in the DSA sequences. Dilated convolution does not enhance the number of convolution kernel parameters. Additionally, it can broaden the receptive field of the network and enhance feature extraction’s capacity for generalization. P3D modules produce higher-level information that is more suited to classifying the stages of MMD, since it not only achieves the same resolution as the input feature, but also has receptive field information that is equivalent to the pooling layer. Atrous convolution is another name for dilated convolution. The convolution kernel is given a fixed number of holes, where the number of holes is equal to the dilated rate r. The size of the dilated convolution kernel is determined using Formula (4):(4)K=r×(k−1)+1
where k is the size of the input convolution kernel, r is the dilated rate, and K is the equivalent convolution kernel size after dilation.

#### 2.3.4. Bottleneck Structure of P3D Module

One can change the dimensions of the feature map and decrease the difficulty of the calculation by adding a 1 × 1 convolution layer before and after the 3 × 3 convolution layer to create a bottleneck structure in ResNet. As shown in Figure 5, three bottleneck structures in P3D module are available when we extend the operation from a 2D CNN to a 3D CNN.

## 3. Results and Discussion

### 3.1. Evaluation Metrics

Accuracy, precision, recall, specificity, F1 score, and AUC are the key evaluation metrics employed in this work. The ratio of samples with the right categorization to all the samples in a multi-classification problem is known as accuracy. Using mild MMD as an example, precision is defined as the ratio of the number of samples accurately recognized as mild MMD to the number of samples predicted to be mild MMD. The proportion of samples that are correctly classified as having mild MMD out of all the samples with mild MMD is known as recall. The ratio of the number of correctly classified non-mild MMD samples to the total number of non-mild MMD samples is known as specificity. The F1 score is the harmonic average of precision and recall. The area under the receiver operating characteristic (ROC) curve is referred to as the AUC. The following is the computation process:(5)$$accuracy=\frac{\sum_{i=1}^{n}{TP}_{i}}{Num}$$
(6)$${precision}_{i}=\frac{{TP}_{i}}{{TP}_{i}+{FP}_{i}}$$
(7)$${recall}_{i}=\frac{{TP}_{i}}{{TP}_{i}+{FN}_{i}}$$
(8)$${specificity}_{i}=\frac{{TN}_{i}}{{FP}_{i}+{TN}_{i}}$$
(9)$${F1\;score}_{i}=2\times \frac{{precision}_{i}\times {recall}_{i}}{{precision}_{i}+{recall}_{i}}$$
(10)$${AUC}_{i}=\frac{\sum_{i\in positiveClass}{rank}_{i}-\frac{M(1+M)}{2}}{M\times N}$$

Among these equations, ${TP}_{i}$ is the number of samples correctly classified as positive in class *i*; ${FP}_{i}$ is the number of samples incorrectly classified as positive in class *i*; ${FN}_{i}$ is the number of samples incorrectly classified as negative in class *i*; ${TN}_{i}$ is the number of samples correctly classified as negative in class *i*; Num is the total number of samples; M is the number of positive samples; N is the number of negative samples; and ${rank}_{i}$ is the serial number.

### 3.2. The Performance of P3D ResNet

The confusion matrix of the model is displayed in Figure 6 in order to examine the effectiveness of the methodology suggested in this paper. Each column’s sum indicates the actual number of samples in this category, whereas each row’s sum represents the number of samples predicted to be this category. It can be observed that the outcome of MMD staging includes the number of accurate classifications and the number predicted to be other categories. The numbers of accurately identified MMD stages are represented by the number on the diagonal path with a deep color. Because of the more pronounced characteristics of severe MMD, in which it is clear that the number of moyamoya-like vessels is greatly decreased, the number of accurately recognized severe MMD samples is the highest. The numbers of correctly classified moderate MMD cases and mild MMD cases are lower than those of severe MMD cases, primarily because mild MMD and moderate MMD may have some similar feature points, making it simple for the model to be interfered with. 

The precision, recall, specificity, F1 score, and AUC of mild, moderate, and severe MMD are all shown in Table 1, in that order. The table shows that this model achieves the maximum precision of 0.971 for severe MMD, and the AUC is 0.99, demonstrating the superior performance of P3D ResNet in the detection of severe MMD. Meanwhile, the precisions for mild and moderate MMD are, respectively, 0.95 and 0.951, and are lower than that for severe MMD. Due to the possibility of misinterpretation between the features of mild MMD and moderate MMD, ROC curves for various MMD stages are displayed in Figure 7. It is clear that the model has the best classification performance for severe MMD because the AUC of the condition is closest to 1. Both mild and moderate MMD have an AUC of 0.96. When calculating the micro-average ROC curve, each component of the label indicator matrix is treated as a label. The macro-average ROC curve is derived from the unweighted mean of each label, and the AUC reaches 0.97, indicating that the model has excellent overall classification performance.

### 3.3. Demonstrations of MMD Staging Based on P3D ResNet

Our findings indicate that P3D ResNet is capable of accurately identifying the MMD stages. The staging results for mild, moderate, and severe cases are shown in Figure 8a–c, respectively. The probabilities are 0.9835, 0.9869, and 0.9901, respectively. In Figure 8a, it can be seen that the ICA and ACA are narrowed at the red arrow, and an abnormal vascular network begins to appear at the skull base, which is defined as mild MMD in this study. In Figure 8b, we can clearly observe that a large number of abnormal vascular networks have formed, which is defined as moderate MMD in this study. Figure 8c shows that the ICAs, ACAs, and a large number of abnormal vascular networks have disappeared, which is defined as severe MMD in this study. It is evident that the model has a positive impact on classifying the MMD stages. This demonstrates the viability and efficacy of the approach in the actual clinical staging of MMD.

### 3.4. Comparison among P3D ResNet Variants

A P3D ResNet, which combined three modules, including P3D-A, P3D-B, and P3D-C, was compared with three different P3D ResNet variations to demonstrate the effectiveness of merging three P3D modules. The P3D-A ResNet was created by substituting the P3D-A modules for all the P3D modules in P3D ResNet. The P3D-B modules were used to replace all the P3D modules to create the P3D-B ResNet, while the P3D-C modules were used to create the P3D-C ResNet.

The accuracy of the P3D ResNet model is 0.0293, 0.026, and 0.0195 higher than that of the three variants, P3D-A ResNet, P3D-B ResNet, and P3D-C ResNet, separately, as can be seen from Table 2. This demonstrates that P3D ResNet is the most effective model due to the diversity of its modules. 

### 3.5. Comparison of P3D ResNet with Different Dilation Rates

The original convolution kernel is represented by a dilation rate of 1. By altering the dilation rate of the network, multiple convolution kernel sizes can be achieved. Therefore, the final performance of the model will be impacted by variable dilation rates. We chose the best dilation rate for training by comparing the accuracy at various dilation rates.

The accuracy of the network increases with the dilation convolution compared to the original model, as shown in Table 3. When the dilation rate is 2, the model’s accuracy reaches its greatest value of 0.9578. However, accuracy starts to suffer as the dilation rate continues to rise. Hefty computing costs, brought on by growing convolution kernel sizes under the influence of dilation rate, are adverse to the increase in model depth and decrease the performance of P3D ResNet. This demonstrates that the model can only be trained well by choosing the appropriate dilation rate. Therefore, 2 was used as the dilation rate in order to guarantee the model’s training effect.

### 3.6. Comparison with Other Models

Three 3D CNN models were chosen for comparison, to demonstrate the superiority of the model suggested in this paper. These models (C3D [25], R3D [26], and R2Plus1D [27]) are frequently employed in video behavior identification, and have produced positive results. The complexity of the model affects how long it takes to train. Excessive parameters will result in a significant rise in the calculation cost and time commitment, which will make it challenging to actually deploy the model and difficult to adapt it to clinical circumstances. We calculated the parameters of P3D ResNet and three other 3D CNN models, and evaluated the classification accuracy of each to validate the performance of P3D ResNet. As shown in Table 4, the R2plus1D and R3D models have the same number of parameters and tiny scales, while their accuracy values are 0.7370 and 0.7922, respectively. The performance of these two models is not sufficient. C3D has more parameters but lower accuracy than P3D ResNet with pretraining. In conclusion, P3D ResNet is more favorable than other models since it can attain high accuracy with fewer parameters. 

The micro-average ROC curve and macro-average ROC curve for each model are displayed in Figure 9a,b, respectively. The AUC of P3D ResNet with pretraining is higher than that of C3D with pretraining, and that of R3D and R2Plus1D without pretraining in both the macro-average and micro-average ROC curves, as can be seen in Figure 8. This demonstrates the effectiveness of the classification effect of P3D ResNet.

## 4. Conclusions

In this paper, a P3D ResNet model is proposed for classifying the stages of MMD. This model can precisely classify MMD stages in the complex vascular network by identifying the features of moyamoya-like vessels and surrounding structures, and could lay a solid foundation for future research. 

The following are the primary contents of this work:

(1) Multiple DSA sequences capable of reflecting blood flow trajectory were chosen, and information on dynamic blood flow was taken into account, which maximized the potential of 3D convolution.

(2) Equivalent 2D convolution and 1D convolution were used to process the spatial and temporal information, respectively, which reduced the scale of the model and improved its capacity for linear expression. The receptive field was broadened by the addition of dilated convolution, and high-dimensional characteristics with richer information were achieved. In order to create P3D modules, 2D dilated convolution and 1D dilated convolution were finally combined through the cascade, parallel, and cascade–parallel modes based on the residual unit. Three different P3D modules were alternately arranged to replace the original residual units in ResNet and form the complete P3D ResNet.

(3) The accuracy of P3D ResNet under various dilation rates was compared to determine the optimum parameters for training. Three variants and three advanced 3D CNN models were compared with P3D ResNet to confirm the efficacy and robustness of P3D ResNet. The experimental findings demonstrate that P3D ResNet, which is superior to its variant and comparative model, has the ability to identify the stages of MMD with an accuracy of 95.78%. It is uncomplicated to deploy in a clinical setting because of the opportune number of parameters and low calculation cost.

The method proposed in this paper still has many aspects that must be improved. The following points can be considered for future improvement:

(1) Data diversification: The data used in this paper are the anterior posterior DSA images of MMD patients’ ICAs. In order to provide more accurate diagnoses of patients in all directions, it is also necessary to refer to other intracranial vessels, such as the external carotid artery and vertebrobasilar artery. At the same time, multimodal images should also be included in the data set to make the diagnosis more comprehensive and reliable. 

(2) Data processing: In this study, the images included in the data set were filtered to remove images with artifacts and unclear development. However, in actual clinical application, there will certainly be artifacts or noise in the DSA images. Therefore, it is necessary to develop data preprocessing algorithms to improve the quality of the input image and improve the final diagnostic accuracy. 

The accuracy of this model’s classification will continue to increase as a result of the increase in MMD samples, advancements in angiography technology, and improvements in CNNs in the future.

## Figures and Tables

**Figure 1 brainsci-13-00742-f001:**
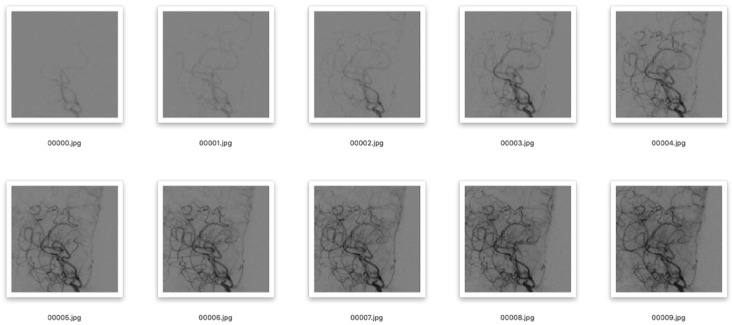
ROI division diagram.

**Figure 2 brainsci-13-00742-f002:**
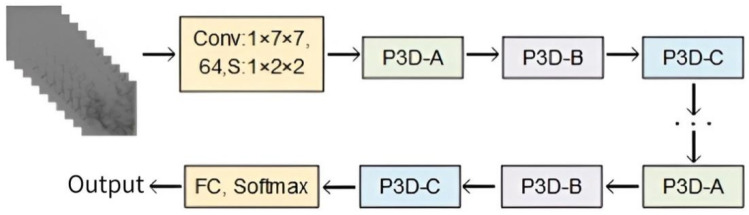
Structure of Pseudo-Three-Dimensional Residual Network (P3D ResNet).

**Figure 3 brainsci-13-00742-f003:**
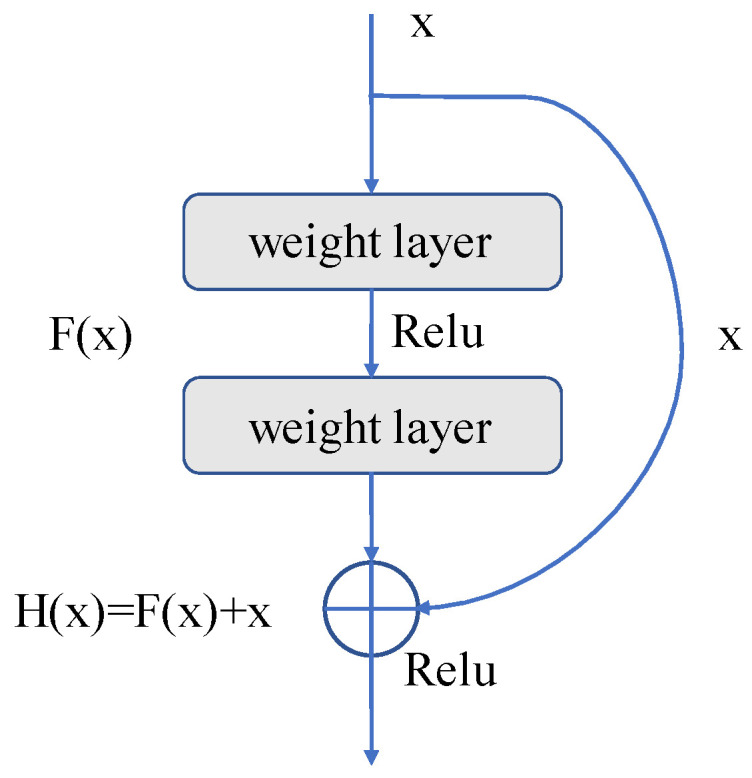
Residual unit.

**Figure 4 brainsci-13-00742-f004:**
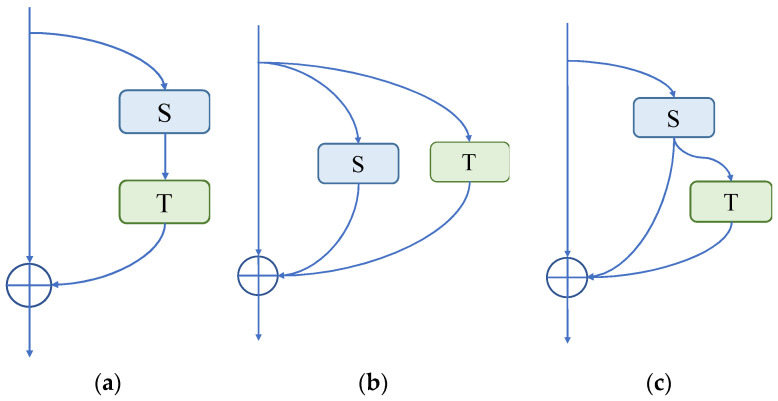
Three different P3D modules: (**a**) P3D-A; (**b**) P3D-B; (**c**) P3D-C. S is defined as the convolution of the spatial dimension, and T is defined as the convolution of the temporal dimension.

**Figure 5 brainsci-13-00742-f005:**
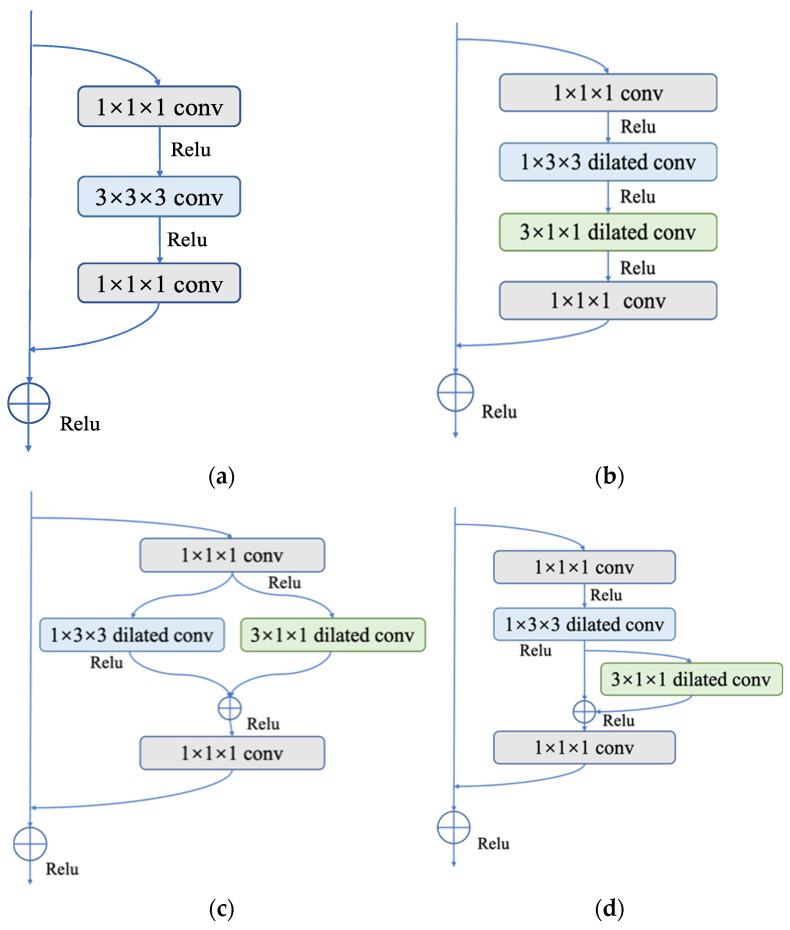
Bottleneck structure of 3D residual element and three different P3D modules. (**a**) Residual unit; (**b**) P3D-A; (**c**) P3D-B; (**d**) P3D-C.

**Figure 6 brainsci-13-00742-f006:**
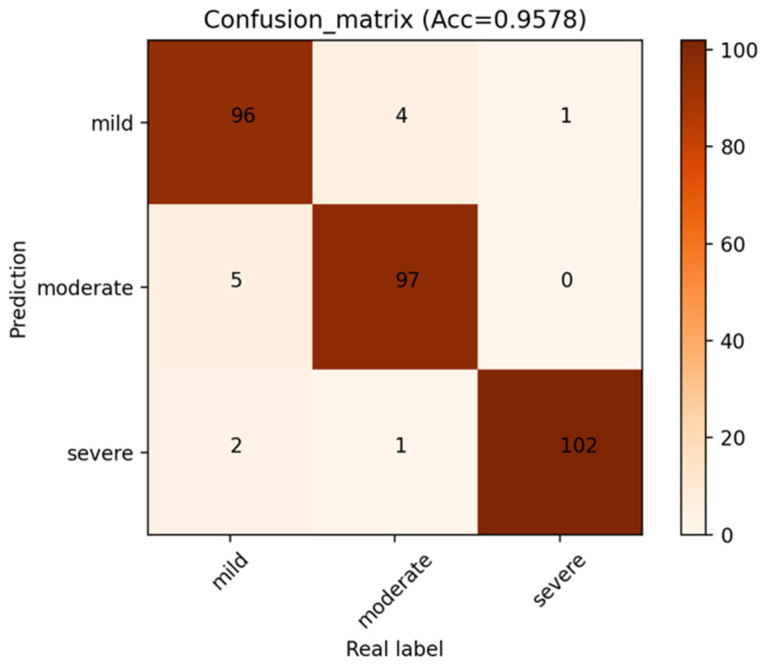
Confusion matrix of P3D ResNet.

**Figure 7 brainsci-13-00742-f007:**
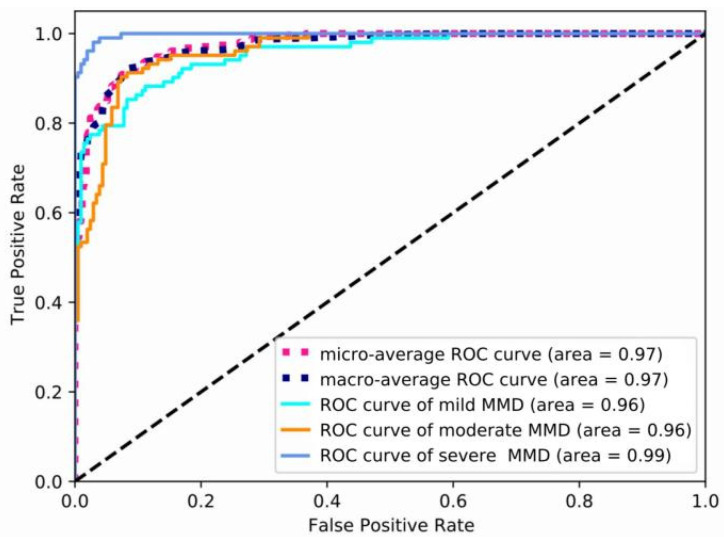
ROC curves of different stages of MMD. MMD represents moyamoya disease, and ROC represents the receiver operating characteristic curve.

**Figure 8 brainsci-13-00742-f008:**
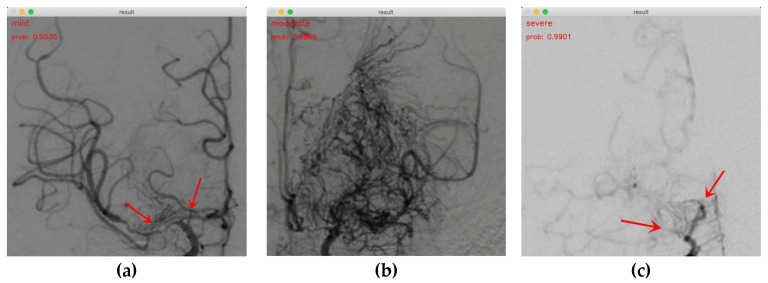
Demonstrations of MMD staging. (**a**) Mild MMD; (**b**) moderate MMD; (**c**) severe MMD.

**Figure 9 brainsci-13-00742-f009:**
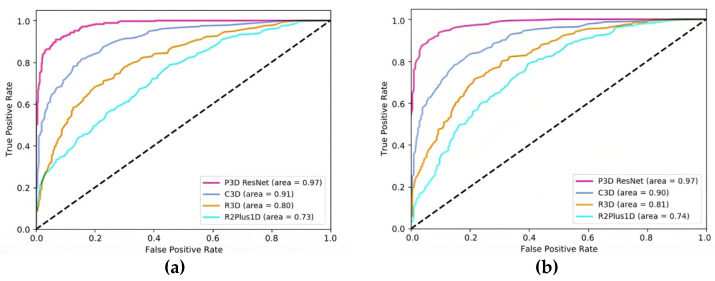
Comparison of ROC curves of different models: (**a**) micro-average ROC curve; (**b**) macro-average ROC curve.

**Table 1 brainsci-13-00742-t001:** MMD staging performance of the model.

Stage	Precision	Recall	Specificity	F1 Score	AUC
Mild	0.95	0.932	0.976	0.941	0.96
Moderate	0.951	0.951	0.976	0.951	0.96
Severe	0.971	0.99	0.985	0.98	0.99

MMD: Moyamoya disease, AUC: the area under ROC.

**Table 2 brainsci-13-00742-t002:** Recognition performance of P3D ResNet and its variants.

Model	Accuracy
P3D-A ResNet	0.9285
P3D-B ResNet	0.9318
P3D-C ResNet	0.9383
P3D ResNet	0.9578

**Table 3 brainsci-13-00742-t003:** Recognition performance of P3D ResNet with different dilation rates.

Dilation Rates	Accuracy
1	0.9448
2	0.9578
3	0.9318
4	0.8961

**Table 4 brainsci-13-00742-t004:** Recognition performance of different models.

Model	Pretraining	Accuracy	Parameters
R2Plus1D	/	0.7370	33.18 M
R3D	/	0.7922	33.18 M
C3D	√	0.8961	78.01 M
P3D ResNet	√	0.9578	65.68 M

## Data Availability

The data presented in this study are available upon request from the corresponding author. The data are not publicly available due to ethical constraints.

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
