# Peer review of "Application of Pseudo-Three-Dimensional Residual Network to Classify the Stages of Moyamoya Disease"

_brainsci, 2023, doi:10.3390/brainsci13050742_

Round 1

Reviewer 1 Report

The paper is very interesting, in line with the advantages of a more objective staging of MMD based on the angiographic pattern.

My contribution is limited to clinical practice considerations only. I hope it can be useful.

It would be helpful to know if there are any potential limitations related to how the DSA data is extrapolated from.

The original sequence of the DSA includes several phases (arterial, capillary, venous, etc.) and each of these has different vascular characteristics; they are also influenced by the operating methods of the DSA study. Could the frame time analysis be influenced by the operational parameters of the DSA study? (standardized flow parameters? Manual or injector injection ? Flow (ml/sec), measurement and catheter placement? Selective injections ? Same level?).

It would be useful to know how much these parameters can contaminate the results and if there are corrective procedures.

It would also be useful to know if a correlation with Perfusion MRI study parameters, which is an integral part of staging in clinical practice, has been investigated, or if there are prospects for it.

About the Reference n. 9 (Kobayashi et Al 2018), it looks like 1978 to me.

Author Response

Dear Editors and Reviewers:

Thank you for your letter and for the reviewers’ comments concerning our manuscript entitled “Apply Pseudo Three Dimensional Residual Network to Classify the Stage of Moyamoya Disease”(brainsci-2237785). Those comments are all valuable and very helpful for revising and improving our paper. We have studied comments carefully and have made correction which we hope meet with approval. The main corrections in the paper and the responds to the reviewer's comments are as following:

  • The reviewer1’s comment:

1.The original sequence of the DSA includes several phases (arterial, capillary, venous, etc.) and each of these has different vascular characteristics; they are also influenced by the operating methods of the DSA study. Could the frame time analysis be influenced by the operational parameters of the DSA study? (standardized flow parameters? Manual or injector injection? Flow (ml/sec), measurement and catheter placement? Selective injections? Same level?). It would be useful to know how much these parameters can contaminate the results and if there are corrective procedures.

The authors’ answer: In this study, DSA images in arterial phase were used, from which the blood flow of arteries could be observed. The condition of patients with MMD is usually clearly displayed in this phase. The relevant operational parameters of DSA will affect the whole length of DSA sequence and time required for the contrast agent to reach the starting point (end of ICA in this study). Since the DSA images required in this study are all manually extracted, the operational parameters have no impact on the research results temporarily. However, if it is necessary to automatically extract DSA images in the future, the operational parameters of DSA must be taken into consideration in order to extract the most appropriate image frame.

2.It would also be useful to know if a correlation with Perfusion MRI study parameters, which is an integral part of staging in clinical practice, has been investigated, or if there are prospects for it.

The authors’ answer: DSA is the first choice for diagnosis for the reason that DSA is gold standard for diagnosis of MMD. Therefore, DSA data is more abundant than other diagnostic methods. As a reliable method for diagnosis of MMD, MRI is also commonly used. But there are fewer patients who do both medical examination at the same time. Therefore, it may be difficult to collect data. We are preparing to incorporate multimodal images into the data set as the future research direction, hoping to provide some help for clinical diagnosis and treatment of MMD.

3.About the Reference n. 9 (Kobayashi et Al 2018), it looks like 1978 to me.

The authors’ answer: We were really sorry for our careless mistakes. Thank you for your remind.

Reviewer 2 Report

The manuscript reports a novel deep-learning based methodology for processing head DSA sequences for automatic determination of MMD severity. The methodology is novel in the sense that a 3D CNN is employed with 2D space and 1D temporal information extracted using novel convolutional pathways included in the design of their CNN.

Nonetheless, the study lacks scientific rigor. The most alarming part of the methodology is the data augmentation done before division of cases into training, validation and testing. This needs to be corrected, and models need to be retrained. If the cases are augmented prior to their division into respective cohorts, there’s a good chance the same case in its augmented form will end up in training, validation, and testing. The results will clearly be an overfit model, that overestimates performance in the testing cohort, as the testing cohort is not truly a testing cohort.

Major comments:

1)    It is unclear from the Introduction, as to why this study is necessary for the 2 following reasons:

a.    Why exactly do we need a CNN for identifying MMD on a 2D DSA sequence? Is it not something that is done relatively quickly already by clinicians? What exactly is the need to automate this process?

b.    Nonetheless, if there does exist such a valiant effort in automatedly predicting the severity of MMD, (Hu et al.) which is performing at >97% in accuracy, sensitivity, and specificity, what exactly is your model going to improve on?

2)    Was there a control cohort? As in a cohort of healthy patients? Are all DSA sequences expected to have Moyamoya disease? Your model has a big assumption that all the inputs have Moyamoya disease, just in different rates of severity.

3)    How were the moderate, mild, and severe MMDs graded? Was this done by experts? If so, was this done by multiple experts to account for the inter-rater variability?

4)    How was the ROI for the pre-processing decided? Any schematic that can clearly indicate the steps out for the readers?

5)    It is frowned upon, and completely unacceptable to perform data augmentation and then divide the cases randomly into training/validation and testing. This allows for the same cases in their augmented forms ending up in training, validation, and testing. IF THAT IS ACTUALLY WHAT WAS DONE, IT NEEDS TO BE CORRECTED, AND MODELS NEED TO BE RETRAINED.

6)    Avoid using non-scientific language like ‘Both the mild and moderate MMD have an AUC of 0.96, which is extremely similar’. It’s either statistically similar or statistically significantly different. The statistical differences between the ROCs can be tested using DeLong’s test. Please employ that to arrive at such conclusions.

7)    In Figure 7, can the authors please point out with arrows, and in their caption using text, as to where the reader should focus to make sense of the DSA images?

Author Response

Dear Editors and Reviewers:
Thank you for your letter and for the reviewers’ comments concerning our manuscript entitled “Apply Pseudo Three Dimensional Residual Network to Classify the Stage of Moyamoya Disease”(brainsci-2237785). Those comments are all valuable and very helpful for revising and improving our paper. We have studied comments carefully and have made correction which we hope meet with approval. The main corrections in the paper and the responds to the reviewer's comments are as following:
The reviewer2’s comment:
a)It is unclear from the Introduction, as to why this study is necessary for the 2 following reasons:
1.    Why exactly do we need a CNN for identifying MMD on a 2D DSA sequence? Is it not something that is done relatively quickly already by clinicians? What exactly is the need to automate this process?
2.    Nonetheless, if there does exist such a valiant effort in automatedly predicting the severity of MMD, (Hu et al.) which is performing at >97% in accuracy, sensitivity, and specificity, what exactly is your model going to improve on?
The authors’ answer: As you said, it is not difficult to quickly diagnose MMD in clinical practice. Usually, clinicians only need to observe the ICA, MCA and ACA of patients to make a diagnosis, but distinguishing the stage of MMD is not just as simple as diagnosis. Clinicians need to carefully observe the state of various structures of blood vessels to make a judgment. The significance of this study is to give clinicians a quick result as a reference for follow-up treatment, which can minimize the risk of sudden cerebrovascular accident for patients with severe illness. Different from the research done by Hu et al., they focus on judging whether patients suffering from MMD. Distinguishing the staging of MMD is the difficulty of the current research. Compared with the traditional 3D convolution, the method we used has less parameters and calculation costs, and has achieved 95.78% accuracy. It is suitable for deployment in clinical, for auxiliary diagnosis, and lays the foundation for realizing the full-process automatic diagnosis of MMD in the future.
b)Was there a control cohort? As in a cohort of healthy patients? Are all DSA sequences expected to have Moyamoya disease? Your model has a big assumption that all the inputs have Moyamoya disease, just in different rates of severity.
The authors’ answer: This study did not use normal samples as the control cohort. All samples were from patients with MMD. Because normal samples can be eliminated easily, our focus is to judge the progress of patients with MMD and provide some reference for follow-up treatment.
c)How were the moderate, mild, and severe MMDs graded? Was this done by experts? If so, was this done by multiple experts to account for the inter-rater variability?
The authors’ answer: All patients in our database were diagnosed independently by two senior neurosurgeons as routine procedures. If a consensus was not reached, the whole treatment team discussed the case together and came to a final consensus. As you are concerned, we have made corrections to our previous draft (2.1. Data processing).  
d) How was the ROI for the pre-processing decided? Any schematic that can clearly indicate the steps out for the readers?
The authors’ answer: We extracted the terminal part of ICA, the MCA and the ACA from the DSA sequence. As is shown in Figure 1, after the ROI was divided, the influence of skull and other irrelevant parts was removed (2.1. Data processing).
e)It is frowned upon, and completely unacceptable to perform data augmentation and then divide the cases randomly into training/validation and testing. This allows for the same cases in their augmented forms ending up in training, validation, and testing. IF THAT IS ACTUALLY WHAT WAS DONE, IT NEEDS TO BE CORRECTED, AND MODELS NEED TO BE RETRAINED.
The authors’ answer: We were really sorry to make you misunderstand because of the unclear expression. In this study, data was not randomly divided into training set, verification set and test set. In view of the imbalance in the number of samples in the three stages of MMD, we focused on data enhancement for mild MMD and severe MMD, while only a small part of the data of moderate MMD has been enhanced. After sufficient data were obtained, and they were balanced in quantity, data enhancement no longer continued. During the process of data enhancement, the enhanced data of the same cases were stored together and finally classified into the training set, verification set and test set respectively. Therefore, the enhanced data of the same cases would not appear in the training set, verification set and test set at the same time. The ratio of 6:2:2 indicates the proportion of the number of each data set to the total number of cases. In addition, we used test time augmentation, that was, to enhance data in test set. The verification of (Simonyan et al.) and (Wang et al.) showed that this method could achieve good results, and this method has also been applied in some competitions, such as ImageNet.
f)Avoid using non-scientific language like ‘Both the mild and moderate MMD have an AUC of 0.96, which is extremely similar’. It’s either statistically similar or statistically significantly different. The statistical differences between the ROCs can be tested using DeLong’s test. Please employ that to arrive at such conclusions.
The authors’ answer: We sincerely thank the reviewer for careful reading. As suggested by the reviewer, we have deleted the sentence with ambiguity and checked whether there was other non-scientific language.
g) In Figure 7, can the authors please point out with arrows, and in their caption using text, as to where the reader should focus to make sense of the DSA images?
The authors’ answer: According to your nice suggestions, we have made corrections to our previous draft, the detailed corrections can be seen in Figure 8. In Figure 8. (a), it can be seen that the ICA and ACA are narrowed at the red arrow, and the abnormal vascular network begins to appear at the skull base, which is defined as mild MMD in this study. In Figure 8. (b), we can clearly observed a large number of abnormal vascular networks have formed, which is defined as moderate MMD in this study. Figure 8. (c) shows that ICA, ACA and a large number of abnormal vascular network have disappeared, which is defined as severe MMD in this study (3.3. Demonstrations of MMD staging based on P3D ResNet).

Reviewer 3 Report

The manuscript “Apply pseudo three-dimensional residual network to clas-sify the stage of moyamoya disease” reports an interesting study on assessing the condition of a rare disease, Moyamoya Disease (MMD) accurately in patients implementing a Pseudo Three-Dimensional Residual Network (P3D ResNet). MMD is quite damaging, with a high mortality rate and disability rate. The experimental results demonstrate that the accuracy of P3D ResNet can reach 95.78% with appropriate parameter quantities, making it easy to implement in a clinical setting. P3D convolution kernels are defined to classify the stage of MMD automatically. These kernels process spatial and temporal data sepa-rately using equivalent 2D convolution and One-Dimensional (1D) convolution.

1.     It is quite interesting to look the design of residual modules. Authors should illustrate the advantages of three different residual modules especially P3D-C.

2.     Statements “Using mild MMD as an example, precision is defined as the ratio of the number of samples accurately recognized as mild MMD to the number of samples predicted to be mild MMD. The proportion of samples that were correctly classified as having mild MMD out of all the samples with mild MMD is known as recall. The ratio of the number of non-mild MMD samples correctly classified to the total number of non-mild MMD samples is known as specificity. The F1 score is the harmonic average of precision and recall. The area under the receiver operating characteristic (ROC) curve is referred to as AUC.” These statements define the work; therefore, I would represent them in bullet points (P3D-C modules were used to create P3D-C ResNet?).

3.     Authors must write about limitations of the model.  

Author Response

Dear Editors and Reviewers:

Thank you for your letter and for the reviewers’ comments concerning our manuscript entitled “Apply Pseudo Three Dimensional Residual Network to Classify the Stage of Moyamoya Disease”(brainsci-2237785). Those comments are all valuable and very helpful for revising and improving our paper. We have studied comments carefully and have made correction which we hope meet with approval. The main corrections in the paper and the responds to the reviewer's comments are as following

The reviewer3’s comment:

  1. It is quite interesting to look the design of residual modules. Authors should illustrate the advantages of three different residual modules especially P3D-C.

The authors’ answer: P3D modules is possible to significantly lower the number of parameters and save calculating cost. In addition, multiple nonlinear operations in the module make it more capable of learning features. Among them, P3D-A connects spatial convolution S and temporal convolution T in series, ensuring the depth of the network under the same receptive field conditions, and improving the performance of the network to a certain extent. P3D-B uses parallel structure to facilitate distributed computing of features. P3D-C integrates series and parallel operations into a module, effectively fusing and supplementing feature information, so as to enrich it. In addition, there are multiple nonlinear operations in these modules, which makes them more capable of learning features. (2.3.2 P3D modules).

  1. Statements “Using mild MMD as an example, precision is defined as the ratio of the number of samples accurately recognized as mild MMD to the number of samples predicted to be mild MMD. The proportion of samples that were correctly classified as having mild MMD out of all the samples with mild MMD is known as recall. The ratio of the number of non-mild MMD samples correctly classified to the total number of non-mild MMD samples is known as specificity. The F1 score is the harmonic average of precision and recall. The area under the receiver operating characteristic (ROC) curve is referred to as AUC.” These statements define the work; therefore, I would represent them in bullet points (P3D-C modules were used to create P3D-C ResNet?).

The authors’ answer: As you are concerned, We used P3D-C modules to create P3D-C ResNet, while P3D ResNet was combined by all the three modules, including P3D-A, P3D-B and P3D-C. We compared the performance of model between P3D ResNet and three variants in 3.4.

  1. Authors must write about limitations of the model.

The authors’ answer: The method proposed in this paper still has many aspects to be improved. The following points can be considered for future improvement:

(1) Data diversification: The data used in this paper is the anterior posterior DSA images of the MMD patients’ ICA. In order to make a more accurate diagnosis of patients in all directions, it is also necessary to refer to other intracranial vessels, such as the external carotid artery and vertebrobasilar artery. At the same time, multimodal images should also be collected into the data set to make the diagnosis more comprehensive and reliable.

(2) Data processing: In this study, the images included in the data set are filtered to remove the images with artifacts and unclear development. However, in the actual clinical application, there will certainly be artifacts or noise in the DSA image. Therefore, it is necessary to develop data preprocessing algorithms to improve the quality of the input image to improve the final diagnostic accuracy. (4. Conclusion).

We tried our best to improve the manuscript and made some changes marked in red in revised paper which will not influence the content and framework of the paper. We appreciate for Editors/Reviewers' warm work earnestly, and hope the correction will meet with approval. Once again, thank you very much for your comments and suggestions.

Round 2

Reviewer 3 Report

This revised version of the manuscript responses reviewer's comments satisfactorily.